# An Anthranilate Derivative Inhibits Glutamate Release and Glutamate Excitotoxicity in Rats

**DOI:** 10.3390/ijms23052641

**Published:** 2022-02-27

**Authors:** Cheng-Wei Lu, Chen-Jung Lin, Pei-Wen Hsieh, Kuan-Ming Chiu, Ming-Yi Lee, Tzu-Yu Lin, Su-Jane Wang

**Affiliations:** 1Department of Anesthesiology, Far-Eastern Memorial Hospital, New Taipei City 22060, Taiwan; drluchengwei@gmail.com (C.-W.L.); lincj176@yahoo.com.tw (C.-J.L.); 2Department of Mechanical Engineering, Yuan Ze University, Taoyuan 32003, Taiwan; 3Research Center for Chinese Herbal Medicine, College of Human Ecology, Chang Gung University of Science and Technology, Taoyuan 33303, Taiwan; pewehs@mail.cgu.edu.tw; 4Graduate Institute of Natural Products, Graduate Institute of Biomedical Sciences, School of Traditional Chinese Medicine, College of Medicine, Chang Gung University, Taoyuan 33303, Taiwan; 5Department of Anesthesiology, Chang Gung Memorial Hospital, Taoyuan 33305, Taiwan; 6Cardiovascular Center, Division of Cardiovascular Surgery, Far-Eastern Memorial Hospital, New Taipei City 22060, Taiwan; chiu9101018@gmail.com (K.-M.C.); mingyi.lee@gmail.com (M.-Y.L.); 7Department of Nursing, Asia Eastern University of Science and Technology, New Taipei City 22060, Taiwan; 8Department of Photonics Engineering, Yuan Ze University, Taoyuan 32003, Taiwan; 9School of Medicine, Fu Jen Catholic University, New Taipei City 24205, Taiwan

**Keywords:** anthranilate derivative, glutamate release, glutamate excitotoxicity, neuroprotection, synaptosomes, kainic acid

## Abstract

The neurotransmitter glutamate plays an essential role in excitatory neurotransmission; however, excessive amounts of glutamate lead to excitotoxicity, which is the most common pathogenic feature of numerous brain disorders. This study aimed to investigate the role of butyl 2-[2-(2-fluorophenyl)acetamido]benzoate (HFP034), a synthesized anthranilate derivative, in the central glutamatergic system. We used rat cerebro-cortical synaptosomes to examine the effect of HFP034 on glutamate release. In addition, we used a rat model of kainic acid (KA)-induced glutamate excitotoxicity to evaluate the neuroprotective potential of HFP034. We showed that HFP034 inhibits 4-aminopyridine (4-AP)-induced glutamate release from synaptosomes, and this inhibition was absent in the absence of extracellular calcium. HFP034-mediated inhibition of glutamate release was associated with decreased 4-AP-evoked Ca^2+^ level elevation and had no effect on synaptosomal membrane potential. The inhibitory effect of HFP034 on evoked glutamate release was suppressed by blocking P/Q-type Ca^2+^ channels and protein kinase C (PKC). Furthermore, HFP034 inhibited the phosphorylation of PKC and its substrate, myristoylated alanine-rich C kinase substrate (MARCKS) in synaptosomes. We also observed that HFP034 pretreatment reduced neuronal death, glutamate concentration, glial activation, and the levels of endoplasmic reticulum stress-related proteins, calpains, glucose-regulated protein 78 (GRP 78), C/EBP homologous protein (CHOP), and caspase-12 in the hippocampus of KA-injected rats. We conclude that HFP034 is a neuroprotective agent that prevents glutamate excitotoxicity, and we suggest that this effect involves inhibition of presynaptic glutamate release through the suppression of P/Q-type Ca^2+^ channels and PKC/MARCKS pathways.

## 1. Introduction

Glutamate is a key excitatory neurotransmitter in brain development, synaptic transmission, synaptic plasticity, and learning and memory processes [1,2]. However, excess glutamate leads to excessive activation of glutamate receptors, which increases the concentration of intracellular calcium and results in the activation of proteases, production of free radicals, induction of mitochondrial dysfunction, and activation of proapoptotic factors. These effects ultimately lead to neurodegeneration and neuronal death. Indeed, glutamatergic excitotoxicity is the most common pathogenic feature of many brain disorders, such as ischemia, epilepsy, and psychiatric and neurodegenerative diseases [3,4]. Therefore, modulating glutamate levels may be a valuable strategy for reducing neurotoxicity and protecting the brain.

Anthranilate derivatives have gained much attention in the area of drug discovery and development because of their diverse pharmacological activities, including anti-inflammatory, antiviral, immunosuppressive, anticancer, antithrombotic, antidiabetic, and analgesic activities [5,6,7,8,9,10,11,12,13,14]. Anthranilate derivatives were also reported to have antidepressant and anticonvulsant effects in different animal models [14,15], indicating that they are potential candidates for drug therapy for related diseases such as anxiety, depression, and epilepsy. Butyl 2-(2-(2-fluorophenyl)acetamido)benzoate (HFP034 is a synthetic anthranilate derivative that is reported to have anti-inflammatory effects [16,17]. However, no studies on the role of HFP034 in the central nervous system (CNS), especially regarding synaptic glutamate release, are available. Therefore, this study aimed to investigate the effect of HFP034 on glutamate release from nerve terminals in the rat cerebral cortex and evaluate its neuroprotective effect in a rat model of excitotoxicity induced by systemic administration of kainic acid (KA), a glutamate analog [18].

## 2. Results

### 2.1. HFP034 Decreases the Release of Glutamate Evoked by 4-AP at Rat Cerebrocortical Nerve Terminals

To analyze whether HFP034 affected synaptic glutamate release, isolated nerve terminals (synaptosomes) were stimulated with 4-aminopyridine (4-AP) (1 mM), which led to the activation of voltage-dependent Ca^2+^ channels (VDCCs) and glutamate release [19]. Under control conditions, the 4-AP (1 mM)-evoked glutamate release from synaptosomes incubated in the presence of 1 mM CaCl_2_ was 6.9 ± 0.1 nmol/mg/5 min. Compared to the control group, in synaptosomes preincubated with HFP034 (10 μM) for 10 min, we observed a significant decrease in 4-AP-evoked glutamate release to 3.6 ± 0.1 nmol/mg/5 min (*n* = 5, *p* < 0.001), whereas no significant change was observed in the basal release of glutamate (*n* = 5, *p* > 0.05). The inhibitory effect of HFP034 on 4-AP-evoked glutamate release was concentration dependent and produced an IC_50_ value of ~6 μM (Figure 1C). Given the robust repression of evoked glutamate release that was seen with 10 μM HFP034, this concentration of HFP034 was used in subsequent experiments to evaluate the mechanisms that underlie the ability of HFP034 to reduce glutamate release.

In addition, we analyzed glutamate release by adding 0.3 mM EGTA to synaptosomes (incubated in the absence of external Ca^2+^) prior to depolarization with 4-AP. This cytosolic release of glutamate amounted to less than 3 nmol/mg/5 min (*p* < 0.001). Notably, this component of release was not affected by the addition of HFP034 (10 μM) (*n* = 5, *p* = 0.92; Figure 1C), suggesting that the inhibition of glutamate release by HFP034 affects the Ca^2+^-dependent exocytotic component of 4-AP-evoked glutamate release. To support this hypothesis, we further analyzed the effects of HFP034 in the presence of dl-threo-β-benzyloxy-aspartate (dl-TBOA), a nonselective inhibitor of all excitatory amino acid transporter subtypes, or bafilomycin A1, a vacuolar H^+^ ATPase inhibitor that causes the depletion of glutamate in synaptic vesicles. As shown in Figure 1C, dl-TBOA (10 μM) increased 4-AP-evoked glutamate release (*p* < 0.001). In the presence of dl-TBOA, HFP034 (10 μM) still effectively decreased 4-AP-evoked glutamate release (*n* = 5, *p* < 0.001). In contrast, bafilomycin A1 (0.1 μM) decreased 4-AP-evoked glutamate release (*p* < 0.001). However, the release measured in the presence of both bafilomycin A1 and HFP034 was similar to that obtained in the presence of bafilomycin A1 alone (*n* = 5, *p* = 0.94). Furthermore, 15 mM KCl-evoked glutamate release, a process that involves Ca^2+^ influx primarily through VDCCs [20], was also decreased by HFP034 (10 μM) (*n* = 5, *p* < 0.001; Figure 1C).

### 2.2. HFP034 Reduces the 4-AP-Induced Increase in (Ca^2+^)_i_ and Does Not Affect Synaptosomal Membrane Potential

Figure 2A shows that the addition of 4-AP (1 mM) increased intra-terminal Ca^2+^ levels ((Ca^2+^)_i_) from 148.3 ± 1.2 nM to a plateau of 224.4 ± 3.2 nM. Preincubation of synaptosomes with HFP034 (10 μM) did not significantly affect basal Ca^2+^ levels (149.5 ± 2.1 nM, *p* = 1) but reduced the 4AP-evoked (Ca^2+^)_C_ increase by 17% (186.6 ± 4.1 nM; *n* = 5, *p* < 0.01). In addition, we used the membrane potential-sensitive dye 3,3,3-dipropylthiadicarbocyanine iodide (DiSC_3_(5)) to assess the effect of HFP034 on the synaptosomal plasma membrane potential. As shown in Figure 2B, the addition of 4-AP (1 mM) caused an increase in DiSC_3_(5) fluorescence in synaptosomes from 0.53 ± 0.3 to 29.3 ± 1.6 fluorescence units/5 min. Preincubation of synaptosomes with HFP034 (10 μM) did not alter the basal DiSC_3_(5) fluorescence (0.55 ± 0.2 fluorescence units/5 min) and produced no substantial change in the 4-AP-mediated increase in DiSC_3_(5) fluorescence (29.5 ± 1.9 fluorescence units/5 min; *p* = 0.93).

### 2.3. HFP034 Decreases a Component of the P/Q-Type Ca^2+^ Channel-Coupled Glutamate Release Pathway

At rat cerebro-cortical nerve terminals, the 4-AP-evoked glutamate release is supported by Ca^2+^ entry through N and P/Q-type VDCCs and Ca^2+^ release from intracellular Ca^2+^ stores [21,22]. Therefore, we assessed which part of the Ca^2+^ source contributed to the effect of HFP034 on 4-AP-evoked glutamate release. As shown in Figure 3, control 4-AP-evoked glutamate release was reduced by 10 μM HFP034 (*n* = 5, *p* < 0.001). Likewise, the blocker of N-type Ca^2+^ channels ω-conotoxin GVIA (ω-CgTX GVIA) (2 µM), the blocker of P/Q-type Ca^2+^ channels ω-agatoxin IVA (ω-Aga IVA) (0.5 µM), the inhibitor of intracellular Ca^2+^ release from the endoplasmic reticulum dantrolene (10 µM), and the inhibitor of mitochondrial Na^+^/Ca^2+^ exchange 7-chloro-5-(2-chlorophenyl)-1,5-dihydro-4,1-benzothiazepin-2(3H)-one (CGP37157) (10 µM) also reduced 4-AP-evoked glutamate release (*p* < 0.001). However, HFP034 (10 µM) could still effectively reduce release in the presence of ω-CgTX GVIA, dantrolene or CGP37157 (*n* = 5, *p* < 0.001). In contrast, further inhibition by HFP034 was not observed in the presence of ω-Aga IVA (*n* = 5, *p* = 0.83). These results suggest the involvement of decreased P/Q-type Ca^2+^ channel activity in the action of HFP034.

### 2.4. The Suppressed Protein Kinase C Pathway Is Linked to HFP034-Mediated Inhibition of Glutamate Release

In view of the demonstrated role of protein kinase C (PKC) in presynaptic modulation [23], we assessed the effect of HFP034 on glutamate release in the presence of the selective PKC inhibitor bisindolylmaleimide I (GF109203X) (10 μM). As shown in Figure 4A, 4-AP (1 mM)-evoked glutamate release was decreased in the presence of GF109203X (10 μM) (*p* < 0.001). HFP034 (10 μM) alone reduced the 4-AP-evoked release of glutamate (*p* < 0.001), but this inhibition was completely suppressed by pretreatment with GF109203X. There was no significant difference between the release after GF109203X alone, and that after GF109203X + HFP034 treatment (*n* = 5, *p* = 0.98). In addition, Figure 4B shows the effects of HFP034 on the phosphorylation of PKC, PKCα and MARCKS (an important presynaptic substrate for PKC) in cerebro-cortical synaptosomes. Compared to the control, 1 mM 4-AP treatment induced a significant increase in the phosphorylation of PKC, PKCα, and MARCKS induced by (*p* < 0.001). When synaptosomes were pretreated with HFP034 (10 μM) for 10 min before the addition of 4-AP, 4-AP-increased phosphorylation of PKC, PKCα, and MARCKS was markedly decreased compared with that in the 4-AP group (*n* = 5, *p* < 0.05; Figure 4B).

### 2.5. HFP034 Attenuates Neuronal Death and Glutamate Elevation in the Hippocampus of Rats with KA

To evaluate whether HFP034 exerted a protective action in a rat model of KA-induced glutamate excitotoxicity, rats were treated with an intraperitoneal (i.p.) injection of HFP034 or dimethyl-sulfoxide (DMSO) for 30 min prior to i.p. injection of KA (15 mg/kg). As shown in Figure 5, we performed Nissl and NeuN staining of brain tissue sections 72 h after KA treatment. Nissl staining revealed significant neuronal loss in the CA1 and CA3 hippocampal regions of the KA-injected rats compared with the DMSO-treated rats (control) (*p* < 0.001). In contrast, Nissl staining in HFP034 + KA-treated rats demonstrated a significant level of protection against hippocampal cell death. Neuronal survival in HFP034 + KA-treated rats was significantly higher than that observed in KA-treated rats (*n* = 3, *p* < 0.05; Figure 5B). The NeuN findings were consistent with those of Nissl staining. Compared to the control group, the KA-treated group experienced a significant decrease in the number of NeuN-positive neurons in the CA1 and CA3 regions (*p* < 0.001). This reduction was prevented in HFP034 + KA-treated rats compared to the KA group (*n* = 3, *p* < 0.05; Figure 5C). In addition, we examined the effect of HFP034 on the concentration of glutamate in the hippocampus of rats with KA (Figure 5D). Compared to the control group, the KA group had significantly increased glutamate levels in the hippocampus (*p* < 0.001). However, groups that received KA and were pretreated with HFP034 experienced a decrease in glutamate levels in the hippocampus, which was significantly different from the KA group (*n* = 3, *p* < 0.05).

### 2.6. HFP034 Reduces the Levels of Endoplasmic Reticulum (ER) Stress-Related Proteins in the Hippocampus of Rats with KA

To investigate how HFP034 attenuated KA-induced neuronal death, we examined whether pretreatment with HFP034 contributed to a reduction in KA-induced ER stress, resulting in neuronal damage [24]. As shown in Figure 6, the levels of ER stress signature molecules, including calpains, glucose-regulated protein 78 (GRP 78), C/EBP homologous protein (CHOP), and caspase-12, were increased in the hippocampus of rats treated with KA compared to the control (*p* < 0.001), but the level of increase decreased with HFP034 pretreatment (*n* = 3, *p* < 0.05 vs. KA group).

### 2.7. HFP034 Suppresses the Activation of Microglia and Astrocytes in the Hippocampus of Rats with KA

Microglial and astrocyte activation is a common pathological feature following KA-induced excitotoxic injury [25]. We examined the effect of HFP034 on KA-induced microglia and astrocyte activation in hippocampal sections using anti-OX42 and anti-GFAP antibodies, respectively. As shown in Figure 7A, we observed a significant increase in the number of OX42^+^ microglial cells in the CA1 and CA3 hippocampal regions of the KA group compared with the control group (*p* < 0.001). In contrast, HFP034-pretreated rats displayed much less OX42 staining, indicating a reduced microglial response to KA-induced injury. Quantification of the results showed a significant decrease in the number of OX42^+^ cells (*n* = 3, *p* < 0.05 vs. KA group; Figure 7B). In addition, the evaluation of GFAP staining for astrocyte activation showed a considerable response in KA-treated rats compared with those in the control group (*p* < 0.001; Figure 7A). Similarly, HFP034 pretreatment significantly reduced astrocyte activation induced by KA, as observed by the reduction in the number of GFAP^+^ cells in the CA1 and CA3 regions (*n* = 3, *p* < 0.05 vs. KA group; Figure 7C).

## 3. Discussion

Excessive release and accumulation of glutamate in the brain is associated with excitotoxicity, a key mechanism in neuronal degeneration in several acute and chronic CNS diseases [3,26]. Identifying new drugs that regulate glutamate release and provide protection against glutamate excitotoxicity is therefore crucial [4,27]. In the current study, we investigated the effect of the anthranilate derivative HFP034 on glutamate release in vitro and its neuroprotective potential against KA-induced glutamate excitotoxicity in vivo. We also examined possible mechanisms underlying the effects of HFP034

In the present study, HFP034 inhibited the release of glutamate evoked by exposing synaptosomes to 4-AP. Because 4-AP induces glutamate release from neurons, two mechanisms (exocytosis and carrier-mediated outward transport) are involved. Exocytotic glutamate release is the result of glutamate release from storage vesicles (Ca^2+^-dependent fraction), whereas glutamate transporters transport glutamate from the axoplasmic site via a reduced Na^+^ gradient (Ca^2+^-dependent fraction) [27]. We found that HFP034 reduced 4-AP-evoked glutamate release in the presence of extracellular Ca^2+^; however, it had no effect in the absence of extracellular Ca^2+^. Furthermore, depletion of glutamate in synaptic vesicles by bafilomycin A1 largely prevented the inhibition of release by HFP034; however, the inhibition of release by HFP034 was insensitive to blockade of the glutamate transporter by dl-TBOA. These results indicate that HFP034 decreases the release of glutamate evoked by 4-AP at cerebro-cortical nerve terminals by inhibiting Ca^2+^-dependent vesicular release rather than reversing the operation of glutamate transporters. In addition, we observed that at cerebro-cortical nerve terminals, HFP034 reduced the 4-AP-evoked increase in (Ca^2+^)_i_, while it did not affect 4-AP-mediated depolarization. This indicates that the inhibitory effect of HFP034 on glutamate release is not due to a decrease in synaptosomal excitability. Furthermore, apart from the 4-AP evolving glutamate release, we found that HFP034 inhibited the release of glutamate evoked by KCl. Because 4-AP-evoked glutamate release involves the activation of Na^+^ and Ca^2+^ channels, 15 mM external KCl-evoked glutamate release involves only Ca^2+^ channels [25], which indicates that Ca^2+^ channels are involved in the effect of HFP034 on glutamate release. Consistent with this, HFP034-mediated inhibition of glutamate release was completely abolished by blocking P/Q-type Ca^2+^ channels but not by blocking N-type Ca^2+^ channels or intracellular Ca^2+^ release. Based on these results, we infer that HFP034 inhibits evoked glutamate release by directly reducing P/Q-type Ca^2+^ channel activity rather than by indirectly affecting nerve terminal excitability.

In synaptic terminals, PKC signaling regulates neurotransmitter release. It has been shown that depolarization-stimulated Ca^2+^ influx increases PKC-dependent phosphorylation and glutamate release [28,29]. In the present study, we found that (1) the inhibitory effect of HFP034 on 4-AP-evoked glutamate release was prevented by PKC inhibition; (2) HFP034 significantly decreased 4-AP-induced phosphorylation of PKC and PKCα, a PKC isoform that is present pre-synaptically in the cerebral cortex [30]; and (3) 4-AP-induced phosphorylation of MARCKS, a major presynaptic substrate for PKC, was decreased by HFP034. Although how HFP034 affects P/Q-type Ca^2+^ channels remains to be elucidated, our data suggest that an intracellular cascade involving the suppression of PKC-dependent pathways is linked to inhibition of 4-AP-evoked glutamate release by HFP034.

We also found that HFP034 exerted neuroprotective efficacy in a rat model of KA-induced glutamate excitotoxicity. The KA-induced excitotoxicity model is a well-established model for screening neuroprotective drugs [31]. Consistent with previous studies [32,33,34], we observed that KA administration (15 mg/kg, i.p.) leads to an increase in glutamate and substantial neuronal death in the hippocampus. These KA-induced alterations were significantly counteracted in the HFP034 pretreatment group, indicating that HFP034 exerts anti-excitotoxic and neuroprotective effects. In addition, we observed that KA increased the expression levels of ER stress-associated proteins, including calpain, GRP78, CHOP, and caspase-12 in the hippocampus; these changes were also reduced by HFP034 pretreatment. These results suggest that HFP034 can reduce ER stress, a key factor in neuronal death in numerous neurological diseases [35,36]. In particular, ER stress can cause neuronal death either by triggering ER Ca^2+^ release, resulting in calpain and caspase-12 activation, or by activating CHOP and GRP78-mediated proapoptotic pathways [37,38,39], which have been suggested to be involved in neuronal cell death after KA-induced excitotoxicity [36]. Furthermore, ER stress inhibition can protect against KA-induced hippocampal neuronal damage [40,41]. Thus, suppressed KA-induced ER stress might partly explain the neuroprotective effect of HFP034.

In addition to ER stress, inflammatory responses, including the activation of microglia and astrocytes, are often associated with KA-induced excitotoxic injury. Activated glial cells increase the production of toxic substances, which in turn contribute to the expansion of brain injury and increased neuron loss. This evidence suggests that the control of KA-induced neuroinflammation is vital to protect hippocampal neurons [25,42]. In the present study, we observed that KA substantially increased the number of activated microglia and astrocytes in the hippocampus, and these increases were suppressed by HFP034 pretreatment. Thus, the suppression of neuroinflammation, in addition to ER stress, could be another mechanism explaining the neuroprotection provided by HFP034 against KA-induced glutamate excitotoxicity. Our finding is consistent with that of previous studies that have reported the anti-inflammatory activities of HFP034 [16,17]. Although how HFP034 suppresses glial activation was not explored in this study, Toll-like receptors (TLRs) have been shown to play a critical role in glial cell activation and subsequent hippocampal neuron excitotoxicity [43,44]. Whether HFP034 inhibits the activation of TLRs, thereby reducing glial cell activation and further suppressing KA-induced excitotoxic insults, is a possibility that should be addressed in future studies.

The ability of HFP034 to reduce glutamate release from nerve terminals may partially explain its neuroprotective mechanism against KA-induced excitotoxicity in vivo. This hypothesis is based on the association between KA-induced neurotoxicity and excessive glutamate release [31,45]. Apart from excessive glutamate release, KA also causes glutamate receptor overstimulation. This overstimulation results in calcium elevation and subsequently triggers intracellular cascade reactions, including reactive oxygen species production, lipid peroxidation, ER stress, and mitochondrial dysfunction and inflammation, eventually leading to cell death [18]. On the basis of these considerations, HFP034 might suppress ER stress and neuroinflammation to prevent KA-induced insults to neuronal integrity, which may be associated with the inhibition of released glutamate.

## 4. Materials and Methods

The experimental protocol was approved by the Fu Jen Institutional Animal Care and Utilization Committee (code A11009). The animals were treated in accordance with the Guide for the Care and Use of Laboratory Animals. The minimum number of animals needed to obtain consistent data was employed.

### 4.1. Materials

HFP034 was synthesized by one of the authors (Pei-Wen Hsieh) [16] and dissolved in 1% DMSO. dl-TBOA, bafilomycin A1, dantrolene, CGP37157, and GF109203X were purchased from Tocris (Bristol, UK). DiSC_3_(5) and fura-2-acetoxymethyl ester (Fura-2-AM) were purchased from Thermo (Waltham, MA, USA). ω-CgTX GVIA and ω-Aga IVA were purchased from the Alomone lab (Jerusalem, Israel). 4-AP, DMSO, KA and all other reagents were purchased from Sigma–Aldrich (St. Louis, MO, USA). Adult male Sprague–Dawley rats (*n* = 42, 150–200 g) were purchased from BioLASCO (Taipei, Taiwan).

### 4.2. Synaptosome Preparation

Rats (*n* = 18) were sacrificed via cervical dislocation and the cerebral cortex was removed immediately. The brain tissue was homogenized in 320 mM sucrose solution and centrifuged at 3000× *g* for 10 min. The supernatant was stratified on Percoll discontinuous gradients and centrifuged at 32,500× *g* for 7 min. The synaptosomal fraction was collected and centrifuged for 10 min at 27,000× *g*. The protein concentration was determined using the Bradford assay. Synaptosomes were centrifuged in a final wash to obtain synaptosomal pellets with 0.5 mg protein, as previously described [46,47,48].

### 4.3. Glutamate Release Analysis

For the glutamate release experiments, the synaptosomal pellet (0.5 mg protein) was resuspended in hepes-buffered solution, and glutamate release was assayed by online fluorimetry [49]. CaCl_2_ (1.2 mM), glutamate dehydrogenase (GDH, 50 units/mL) and NADP^+^ (2 mM) were added at the start of the incubation. Glutamate release was induced with 4-AP (1 mM) and monitored by measuring the increase in fluorescence (excitation and emission wavelengths of 340 and 460 nm, respectively) resulting from NADPH being produced by oxidative deamination of released glutamate by GDH. The amount of released glutamate was calibrated against a standard of exogenous glutamate (5 nmol) and expressed as nanomoles glutamate per milligram synaptosomal protein (nmol/mg).

### 4.4. Intrasynaptosomal Ca^2+^ Concentration ([Ca^2+^]_i_)

Synaptosomes (0.5 mg protein) were incubated in hepes-buffered solution containing Fura 2-AM (5 μM) and CaCl_2_ (0.1 mM) for 30 min at 37 °C. Samples were centrifuged for 1 min at 3000× *g*, and pellets were resuspended in hepes-buffered medium containing CaCl_2_ (1.2 mM). Fura-2-Ca fluorescence was monitored at 5 s intervals for 5 min. (Ca^2+^)_i_ (nM) was calculated using previously described calibration procedures and equations [50].

### 4.5. Membrane Potential

The synaptosomal membrane potential was assayed with the positively charged membrane potential-sensitive carbocyanine dye DiSC_3_(5). DiSC_3_(5) fluorescence was monitored at 2 s intervals, and the data are expressed in fluorescence units [51].

### 4.6. Histological Analysis

The rats (*n* = 24) were divided into four experimental groups: the DMSO-treated group (control), KA-treated group, HFP034 10 mg/kg + KA group, and HFP034 30 mg/kg + KA group. HFP034 was dissolved in a saline solution containing 1% DMSO and administered (i.p.) 30 min before KA injection (15 mg/kg in 0.9% NaCl, pH 7.0, i.p.). The dose and schedule of administration were chosen based on previous experiments of our group [34,49] and others [18,25]. For Nissl staining, rats (*n* = 3 per group) were euthanized 72 h after KA injection by trans-cardial perfusion with 4% paraformaldehyde in 0.1 M phosphate-buffered saline (PBS) under inhalational anesthesia with 2–3% iso-furane. The brains were removed, fixed overnight with 4% paraformaldehyde solution, and cryoprotected in sucrose phosphate buffer at 4 °C. The brains were cut into 30 µm coronal sections, mounted on gelatinized slides, air-dried and stained with 0.1% aqueous cresyl violet stain (Sigma Chemicals, St. Louis, MO, USA) for 20 min. Then, the slides were washed in distilled water, differentiated in 70% ethyl alcohol, dehydrated in ascending grades of ethyl alcohol, cleared in xylene, and mounted with DPX (Sigma Chemicals, St. Louis, MO, USA). For immunofluorescence staining, the brain sections were blocked with 2% bovine serum albumin (BSA) in PBS for 30 min and then incubated overnight at 4 °C with the primary antibodies anti-NeuN (1:500, Abcam, Cambridge, UK), anti-OX42 (1:500, Merck Millipore, Burlington, VT, USA), and anti-GFAP (1:1000, Cell Signaling, MA, USA). The sections were incubated for 90 min at room temperature with the corresponding secondary antibodies (1:1000, Alexa Fluor 488, DyLight 594, Invitrogen, CA, USA), mounted on gelatin-coated slides and cover-slipped with VectaShield medium (Vector Labs, Burlingame, CA, USA). Cells were stained with the nuclear staining dye DAPI (1 µg/mL, Sigma–Aldrich) for 20 s. Images were captured with an upright fluorescence microscope (Zeiss Axioskop 40, Goettingen, Germany) using ×4 (aperture is 0.1) and ×10 (aperture is 0.25) objectives. The numbers of living neurons and NeuN^+^, OX42^+^, and GFAP^+^ cells were counted in a 255 μm × 255 μm area of the hippocampal CA1 and CA3 using ImageJ software (Synoptics, Cambridge, UK).

### 4.7. High-Performance Liquid Chromatography

Determination of glutamate concentrations in brain tissue was performed with a high-performance liquid chromatography (HPLC) system with electrochemical detection (HTEC-500) [34]. Briefly, the rats were sacrificed through decapitation at 72 h after KA injection and the brains were removed immediately. The hippocampi were dissected and homogenized in 5 mL of hepes-buffered medium. The homogenate was centrifuged at 1500× *g* at 4 °C for 10 min, and then the supernatant (10 μL) was filtered through 0.22 µm filters before injection into the HPLC system. An Eicompak GU-GEL column, a glutamate oxidase-immobilized column (EENZYMPAK), and a platinum electrode set at 450 mV against an Ag/AgCl reference electrode were used. The mobile phase comprised 50 mM NH_4_Cl and 250 mg/L hexadecyltrimethylammonium bromide with pH set at 7.4. The column temperature was 35 °C, and the flow rate was 1.2 mL/min. The relative free glutamate concentration was determined using peak areas with an external standard method. Serial dilutions of the standards were injected, and their peak areas were determined. A linear standard curve was constructed by plotting the peak areas versus corresponding concentrations of each standard.

### 4.8. Western Blot

Synaptosomes and hippocampal tissue were lysed in ice-cold Tris–HCl buffer solution and centrifuged for 10 min at 13,000× g at 4 °C. The protein concentration in the supernatant was measured using the Bradford protein assay (Bio–Rad laboratories, Hercules, CA, USA). Equal amounts (30 µg) of protein were loaded into each lane on a 10% polyacrylamide gel and then transferred to a polyvinylidene-difluoride (PVDF) membrane in a semidry system (Bio–Rad, Hercules, CA, USA) for 120 min. Transferred membranes were blocked for 1 h in 5% nonfat dry milk in TBST (25 mM Tris-HCl, pH 7.5, 125 mM NaCl, and 0.05% Tween 20) and incubated overnight at 4 °C with specific primary antibodies [anti-PKC, 1:700, Abcam, Cambridge, UK); phospho-PKC (1:2000, Cell Signaling, MA, USA); PKCα (1:600, Cell Signaling, MA, USA), phospho-PKCα (1:2000, Abcam, Cambridge, UK); phospho-MARCKS (1:250, Cell Signaling, MA, USA); calpain 1 (1:2000, Abcam, Cambridge, UK), calpain 2 (1:800, Millipore); caspase 12 (1:3000, Abcam, Cambridge, UK); CHOP (1:300, Santa Cruz, TX, USA); GRP 78 (1:1500, Abcam, Cambridge, UK); and β-actin (1:8000, Cell Signaling, MA, USA). Membranes were washed with TBST for 15 min and incubated with horseradish peroxidase-coupled secondary antibodies (1:16,000, GeneTex, CA, USA) for 1 h at room temperature. Then, the specific protein bands were visualized using film exposure with a chemiluminescence detection system (GeneTex, CA, USA) and quantified using ImageJ software (Synoptics, Cambridge, UK).

### 4.9. Statistical Analyses

The results are expressed as the mean ± standard error of the mean (S.E.M.). Statistical analysis was performed using GraphPad Prism-8 software (GraphPad Inc., San Diego, CA, USA). When testing the significance of the effect of HFP034 versus the control, Student’s t test was used. When comparing the effect of HFP034 in different experimental conditions, one-way analysis of variance (ANOVA) followed by Tukey’s post hoc test was used. *p* < 0.05 was considered to indicate a statistically significant difference between groups.

## 5. Conclusions

In summary, we demonstrated that the anthranilate derivative HFP034 inhibits glutamate release from rat cerebro-cortical nerve terminals by suppressing P/Q-type Ca^2+^ channels and the PKC/MARCKS pathways and that HFP034 prevents KA-induced glutamate neurotoxicity in vivo by inhibiting ER stress and neuroinflammation (Figure 8). To our knowledge, this is the first report to demonstrate the inhibitory role of HFP034 in glutamate release and glutamate excitotoxicity. This finding may provide a pharmacological basis for the clinical use of HFP034 in the treatment of CNS diseases involving glutamate excitotoxicity.

## Figures and Tables

**Figure 1 ijms-23-02641-f001:**
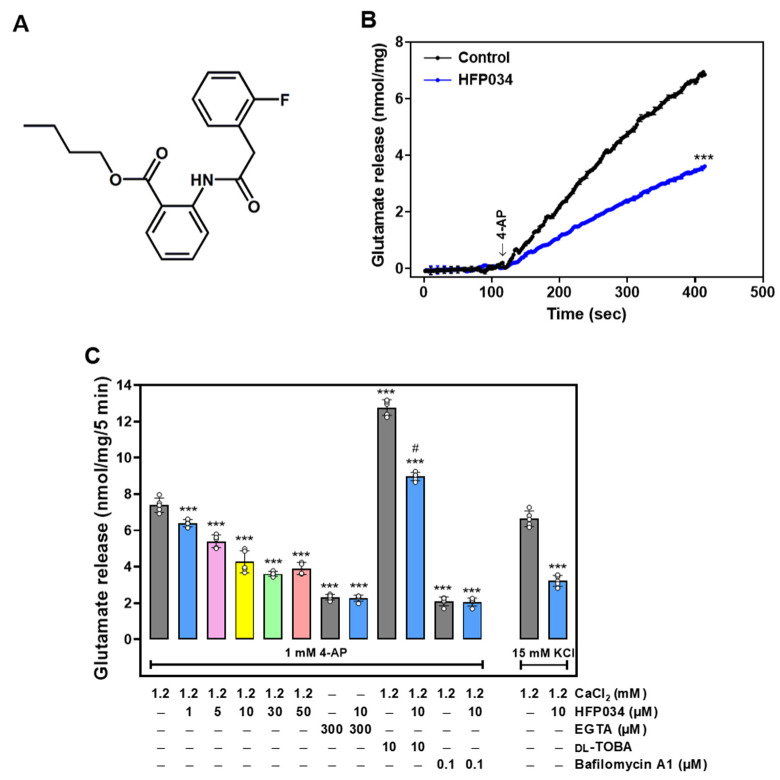
Effect of HFP034 on 4-AP-evoked glutamate release from rat cerebro-cortical nerve terminals. (**A**) The chemical structure of HFP034. (**B**) Glutamate release was measured under control conditions or in the presence of 10 µM HFP034 added 10 min prior to the addition of 4-AP (1 mM). (**C**) Effect of HFP034 at different concentrations on 4-AP-evoked glutamate release, and extracellular Ca^2+^-free solution, glutamate transporter inhibitor dl-TBOA or vesicular glutamate transporter inhibitor bafilomycin A1 on the effect of HFP034. Effect of HFP034 on the release of glutamate evoked by 15 mM KCl is shown in the Figure 1C. Each dot represents the value for an individual experiment. Data are presented as mean ± S.E.M. (*n* = 5 per group). *** *p* < 0.001 vs. control group; # *p* < 0.001 vs. dl-TBOA-treated group.

**Figure 2 ijms-23-02641-f002:**
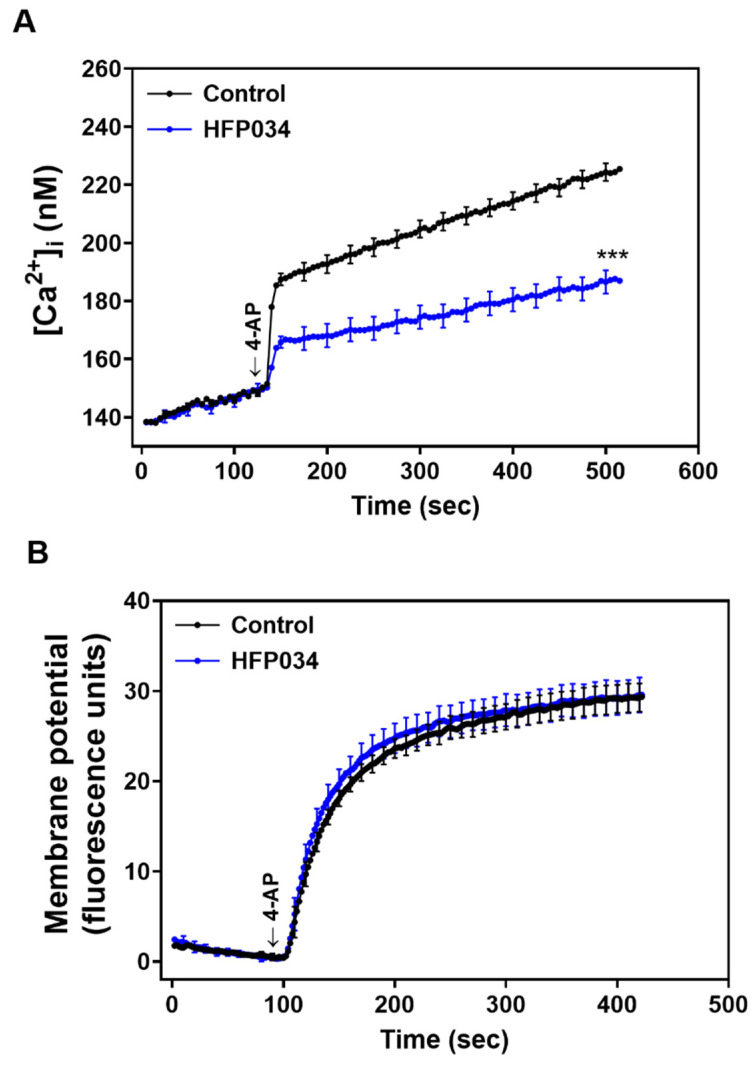
Effect of eupafolin on [Ca^2+^]_i_ (**A**) and the synaptosomal membrane potential (**B**). HFP034 (10 µM) was added 10 min before the addition of 4-AP. Data are presented as mean ± S.E.M. (*n* = 5 per group). *** *p* < 0.001, versus control group.

**Figure 3 ijms-23-02641-f003:**
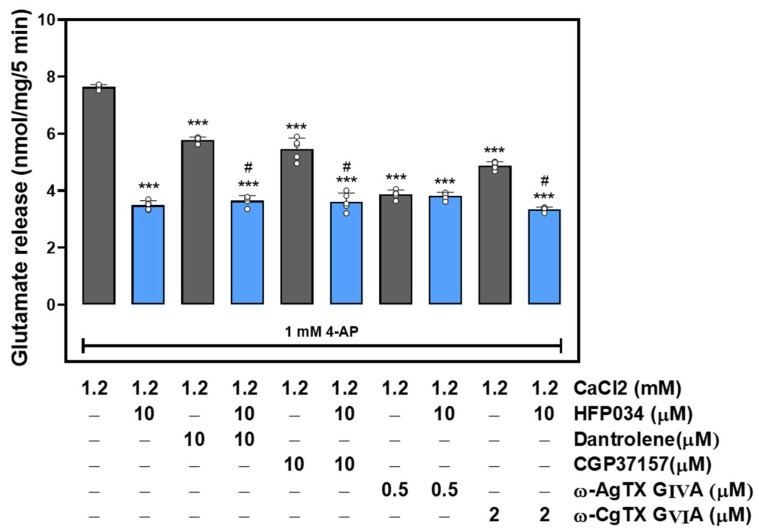
Effect of HFP034 on 4-AP-evoked glutamate release in the presence of N-type Ca^2+^ channel blocker ω-CgTX GVIA, P/Q-type Ca^2+^ channel blocker ω-AgTX IVA, ryanodine receptor inhibitor dantrolene, or mitochondrial Na^+^/Ca^2+^ exchanger inhibitor CGP37157. HFP034 was added 10 min before the addition of 4-AP, and other drugs were added 10 min before this. Each dot represents the value for an individual experiment. Data are presented as mean ± S.E.M. (*n* = 5 per group). *** *p* < 0.001, versus control group. # *p* < 0.001, versus dantrolene- or CGP37157-treated group.

**Figure 4 ijms-23-02641-f004:**
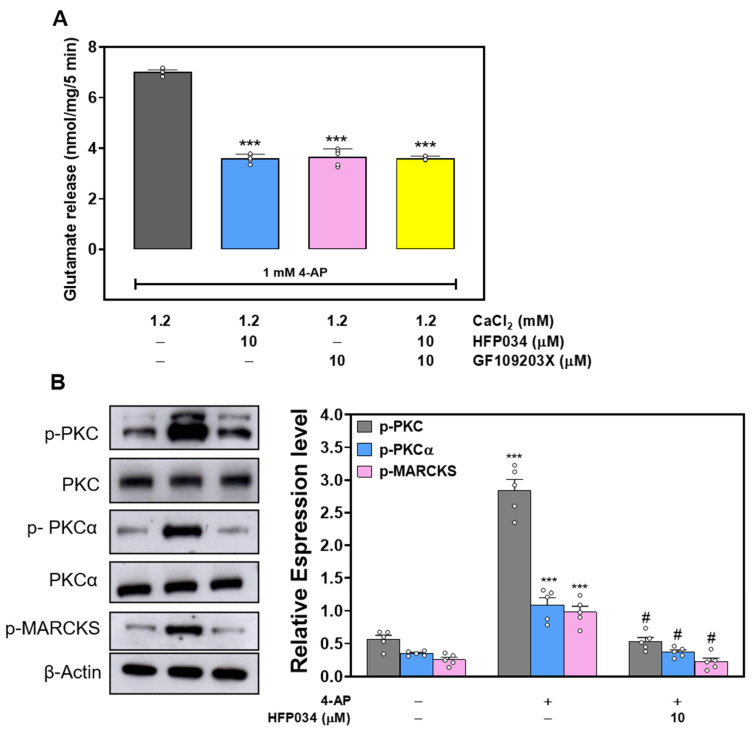
(**A**) Effect of the PKC inhibitor GF109203X on the HFP034-mediated inhibition of 4-AP-evoked glutamate release. (**B**) Effect of HFP034 on PKC and MARCKS phosphorylation evoked by 4-AP. HFP034 or GF109203X was added 10 min before the addition of 4-AP. Each dot represents the value for an individual experiment. Data are presented as mean ± S.E.M. (*n* = 5 per group). *** *p* < 0.001, versus control group. # *p* < 0.001, versus GF109203X- or 4-AP-treated group.

**Figure 5 ijms-23-02641-f005:**
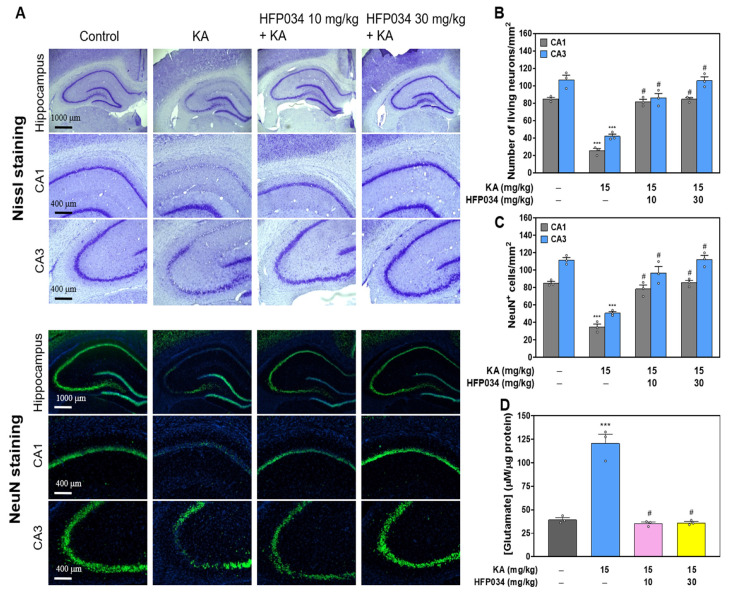
Effect of HFP034 pretreatment on the neuronal cell death and glutamate levels in the hippocampus of rats with KA. (**A**) Representative images of crystal violet and NeuN staining at 3 d after i.p. KA. (**B**,**C**) Quantitative data of A showing the number of living neurons and NeuN^+^ cells in the hippocampal CA1 and CA3 regions. (**D**) The effect of HFP034 on the concentration of glutamate in the hippocampus of rats with KA. Each dot represents the value for an individual experiment. Data are presented as mean ± S.E.M. *n* = 3 rats for each group. *** *p* < 0.001, versus control group. # *p* < 0.001, versus KA group.

**Figure 6 ijms-23-02641-f006:**
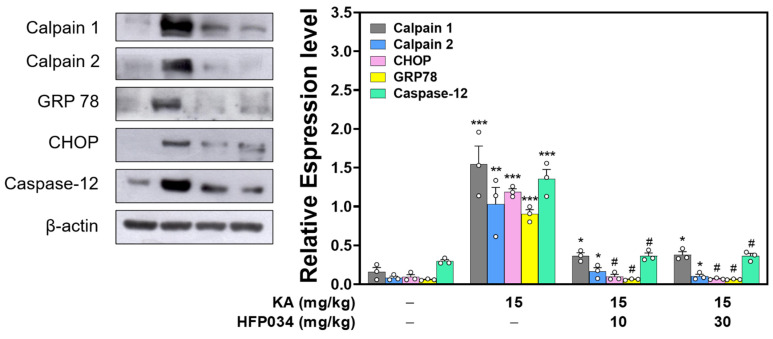
Effect of HFP034 pretreatment on the expression levels of ER stress-associated proteins, calpains, GRP78, CHOP, and caspase-12 in the hippocampus of rats with KA. Each dot represents the value for an individual experiment. Data are presented as mean ± S.E.M. *n* = 3 rats for each group. ** *p* < 0.01, *** *p* < 0.001 versus control group. * *p* < 0.01, # *p* < 0.001, versus KA group.

**Figure 7 ijms-23-02641-f007:**
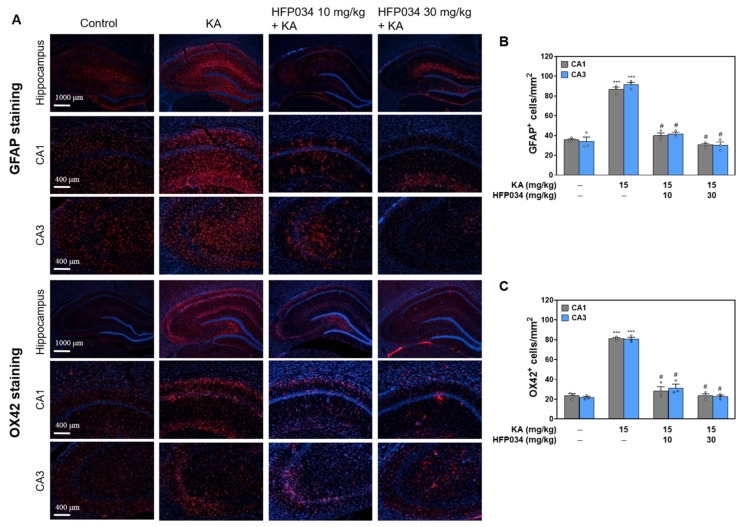
Effect of HFP034 pretreatment on the activation of microglia and astrocyte in the hippocampus of rats with KA. (**A**) Representative images of OX42 and GFAP staining at 3 d after i.p. KA. (**B**,**C**) Quantitative data of A and C showing the number of OX42^+^ and GFAP^+^ cells in the hippocampal CA1 and CA3 regions. Each dot represents the value for an individual experiment. Data are presented as mean ± S.E.M. *n* = 3 rats for each group. *** *p* < 0.001, versus control group. # *p* < 0.001, versus KA group.

**Figure 8 ijms-23-02641-f008:**
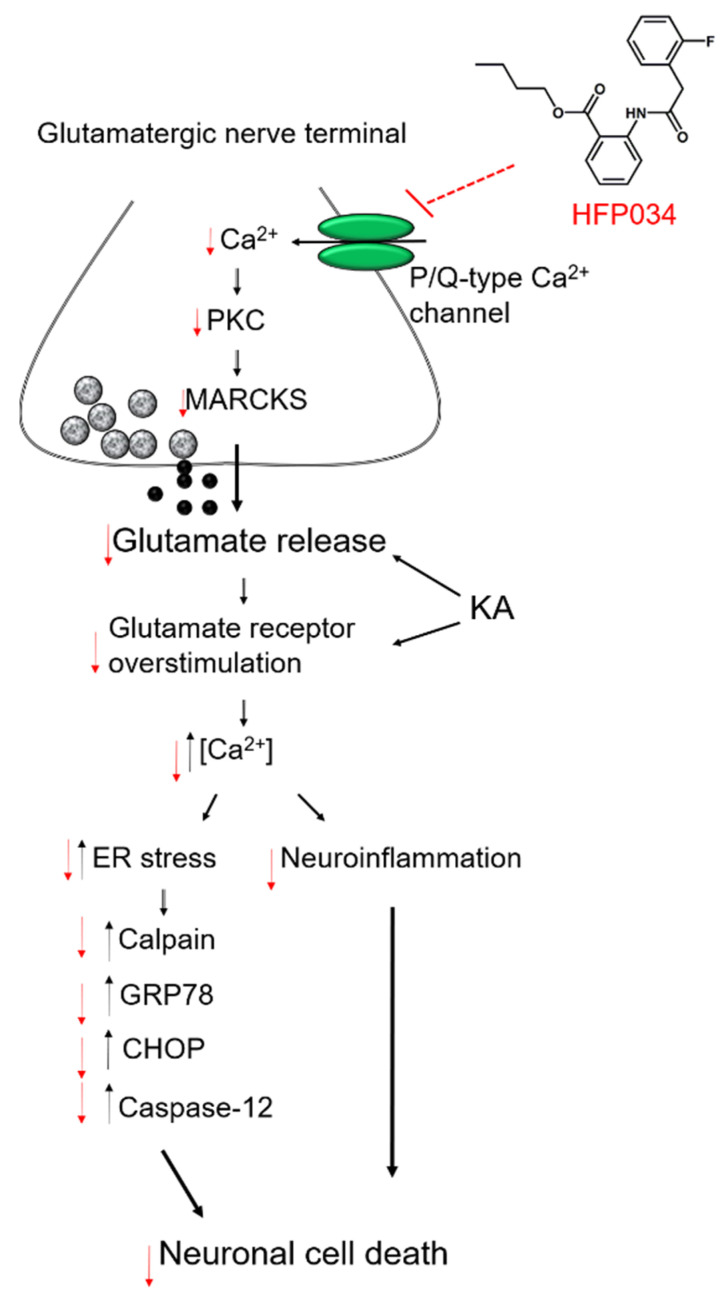
A proposed mechanism underlying the inhibition of glutamate release and glutamate excitotoxicity by HFP034 in rats. HFP034 depresses glutamate release via suppressing presynaptic P/Q-type Ca^2+^ channels and PKC/MARCKS pathway, as well as protecting KA-induced neuronal death via ER stress and neuroinflammation inhibition. Black arrows indicate positive regulation, and red arrows indicate negative regulation.

## Data Availability

The data presented in this study are available on request from the corresponding author.

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
