# Peer review of "An Anthranilate Derivative Inhibits Glutamate Release and Glutamate Excitotoxicity in Rats"

_ijms, 2022, doi:10.3390/ijms23052641_

Round 1

Reviewer 1 Report

The manuscript entitled: An anthranilate derivative inhibits glutamate release and glutamate excitotoxicity in rats by Lu et al. reported the good work. The research presented that derivatives of anthranilate was used to inhibit the release of glutamate from the rat cerebrocortical nerve terminals by suppressing Ca2+ channels and the PKC/MARCKS pathways. Moreover, these derivative prevents the kainic acid induced glutamate neurotoxicity and the investigation were done through inhibiting endoplasmic reticulum stress and neuroinflammation in in-vivo studies. The data is presented well in the form of figures and graphs. Therefore, I recommend the publication in its present form.

Author Response

We thank the reviewer for the critical comments.

Reviewer 2 Report

Interesting and well-written paper. Well documented on the experimental side. The results of the experiments confirm the thesis about the neuroprotection of HFP034. I recommend it for publication. Minor critical remarks: 1. Why did the authors choose a HFP034 concentration of 10 µM or 10 mg/kg? I did not find this information in the text. 2. It would be valuable to include the results of HFP034 neurotoxicity eg in neuronal culture. 3. The methodology for determining glutamate by means of HPLC (4.7) is described very briefly in relation to other experiments. Details are not available in the quoted publication [34]. 4. The chemical structure of the compound suggests very limited solubility in aqueous solutions. If this is the case, a few words of comment on the use of HFP034 would be useful. 

Author Response

We thank the reviewer for the critical comments and constructive suggestions.

Interesting and well-written paper. Well documented on the experimental side. The results of the experiments confirm the thesis about the neuroprotection of HFP034. I recommend it for publication. 

Minor critical remarks: 

  1. Why did the authors choose a HFP034 concentration of 10 µM or 10 mg/kg? I did not find this information in the text.

As suggestion by the reviewer, the sentence Given the robust repression of evoked glutamate release that was seen with 10 mM HFP034, this concentration of HFP034 was used in subsequent experiments to evaluate the mechanisms that underlie the ability of HFP034 to reduce glutamate release.The dose and schedule of administration were chosen based on previous experiments of our group and othersare added in the result section (Page 2, Lines 83-85; Page 12, Lines 374-376).

  1. It would be valuable to include the results of HFP034 neurotoxicity eg in neuronal culture.

The reviewer's recommendation is good. Due to the limitations of the experimental equipment, the reviewer's recommendation could not be completed in the manuscript. We will incorporate this recommendation into future experimental designs. Hope the reviewer will accept this reply.

  1. The methodology for determining glutamate by means of HPLC (4.7) is described very briefly in relation to other experiments.Details are not available in the quoted publication [34].

As suggestion by the reviewer, the sentences are modified to Briefly, the rats were sacrificed through decapitation at 72 h after KA injection and the brains were removed immediately. The hippocampi were dissected and homogenized in 5 ml of hepes-buffered medium.and An Eicompak GU-GEL column, a glutamate oxidase-immobilized column (EENZYMPAK), and a platinum electrode set at 450 mV against an Ag/AgCl reference electrode were used. The mobile phase comprised 50 mM NH4Cl and 250 mg/L hexadecyltrimethylammonium bromide with pH set at 7.4. The column temperature was 35 0C, and the flow rate was 1.2 ml/min. (Page 12, Lines 401-410).

  1. The chemical structure of the compound suggests very limited solubility in aqueous solutions.If this is the case, a few words of comment on the use of HFP034 would be useful.

According to this point, in the method section , the sentence is modified to HFP034 was synthesized by one of the authors (Pei-Wen Hsieh) [16] and dissolved in 1% DMSO(Page 11, Lines 333-334).
